# The seasonality of infections in tropical Far North Queensland, Australia: A 21-year retrospective evaluation of the seasonal patterns of six endemic pathogens

Lee J. Fairhead[1], Simon Smith[1], Beatrice Z. Sim [2‡], Alexandra G. A. Stewart [3‡], James D. Stewart [1‡], Enzo Binotto[1‡], Matthew Law [4‡], Josh Hanson [1,4] *

1 Department of Medicine, Cairns Hospital, Cairns, Queensland, Australia, 2 Infectious Diseases Service, Royal Adelaide Hospital, Adelaide, South Australia, Australia, 3 Infectious Diseases Department, Western Health, Melbourne, Victoria, Australia, 4 The Kirby Institute, University of New South Wales, Sydney, New South Wales, Australia

☯ These authors contributed equally to this work.
‡ BZS, AGAS, JDS, EB and ML also contributed equally to this work.
* jhanson@kirby.unsw.edu.au

## Abstract

An understanding of the seasonality of infections informs public health strategies and assists clinicians in their management of patients with undifferentiated illness. The seasonality of infections is driven by a variety of environmental and human factors; however, the role of individual climatic factors has garnered much attention. This study utilises Poisson regression models to assess the seasonality of six important infections in tropical Australia and their association with climatic factors and severe weather events over a 21-year period. Melioidosis and leptospirosis showed marked wet season predominance, while more cases of rickettsial disease and cryptococcosis were seen in cooler, drier months. *Staphylococcus aureus* infections were not seasonal, while influenza demonstrated inter-seasonality. The climate did not significantly change during the 21 years of the study period, but the incidence of melioidosis and rickettsial disease increased considerably, highlighting the primacy of other factors—including societal inequality, and the impact of urban expansion—in the incidence of these infections. While anthropogenic climate change poses a threat to the region —and may influence the burden of these infections in the future—this study highlights the fact that, even for seasonal diseases, other factors presently have a greater effect on disease incidence. Public health strategies must also target these broader drivers of infection if they are to be effective.

## Introduction

Seasonality—periodic changes in disease incidence which correspond to the seasons or other stereotyped calendar periods—is a characteristic of many infectious diseases [1, 2] Knowledge

**Data Availability Statement:** The data used to prepare this manuscript have been uploaded as a supplementary file (S1 Dataset).

**Funding:** The authors received no specific funding for this work.

**Competing interests:** The authors have declared that no competing interests exist.

of the seasonality of different infections enhances understanding of host-pathogen interactions, informs public health policy and facilitates clinicians' recognition of undifferentiated illness. The seasonality of different infections results from the complex interplay of environmental and human factors, with individual components of the climate and weather events recently garnering much scientific attention [3].

Anthropogenic climate change is expected to impact the distribution and incidence of common infections with tropical regions likely to be disproportionately affected [4, 5]. This is important because life-threatening tropical infections like melioidosis have been linked to rainfall and cloud cover while leptospirosis has been associated with rainfall and flooding events [6, 7]. The seasonality of other infectious diseases encountered commonly in the tropics —including *Staphylococcus aureus*, rickettsial and cryptococcal infections—has also been reported with specific climate elements implicated; particularly rainfall, temperature, and humidity [8–11]. As temperatures rise and severe weather events become more frequent, it is important to have a deeper understanding of the role that individual climate factors play in the incidence of these infections. However, climate is not the only arbiter of disease incidence, there are human and environmental factors which also need to be considered when examining the seasonality of different infections. Human behaviour changes with seasons, with social, occupational and recreational activity often showing marked inter-seasonal variation [1, 12]. The way in which humans interact with the natural environment can also vary by season, resulting in differences in exposure to pathogens or to the vectors that carry them [13, 14].

Far North Queensland (FNQ) in the north-eastern tip of tropical Australia, covers an area of 380,000km$^2$. It extends from the Wet Tropics in the south of the region to the Torres Strait Islands in the north (Fig 1). The region's climate is characterised by perennial humidity, hot, wet summers with a risk of cyclones and flooding and mild, drier winters. Cairns—the region's main city—is a popular international tourist destination but the remainder of FNQ is significantly less developed, and includes 3 of the 10 most socio-economically disadvantaged regions in Australia [15]. This socioeconomic disadvantage is disproportionately borne by the local Indigenous population who represent approximately 17% of the region's nearly 290,000 residents [16].

The infectious diseases encountered in the region are linked to this unique environment. FNQ shares a maritime border with PNG and after the establishment of the Torres Strait treaty in 1985, residents of PNG and the outer Torres Strait Islands have been able to move freely across the border to maintain cultural ties. However, the fragile public health system in PNG increases the risk of imported infectious diseases [17–19]. Infectious diseases can also be introduced by tourists and returning travelers, while the vibrant outdoor lifestyle and diverse local fauna increase the risk of zoonotic infection [20–22]. The significant socioeconomic disadvantage experienced by many local Indigenous people results in a high burden of chronic health conditions, which increases their susceptibility to infection [23–26].

There are several important pathogens that are endemic in this tropical region. The local, thriving farming sector means that agricultural workers are at increased risk of leptospirosis with the infection's local incidence one of the highest in the developed world [27]. Meanwhile, urban expansion and high rates of predisposing comorbidities are associated with a rising local incidence of melioidosis [28]. The rates of *Orientia tsutsugamushi* and *Rickettsia australis* infection (which cause scrub typhus and, Queensland Tick Typhus (QTT) respectively) are also increasing as the urban fringe encroaches on traditionally rural areas [29]. Endemicity of Eucalyptus trees and a significant rural-dwelling population likely increases the risk of *Cryptococcus gattii* infection [30]. There is a significant local burden of *Staphylococcus aureus* infection, with rates of methicillin-resistant *S. aureus* (MRSA) amongst the highest in Australia [31]. Meanwhile, the inter-seasonality of influenza is intensifying [32].

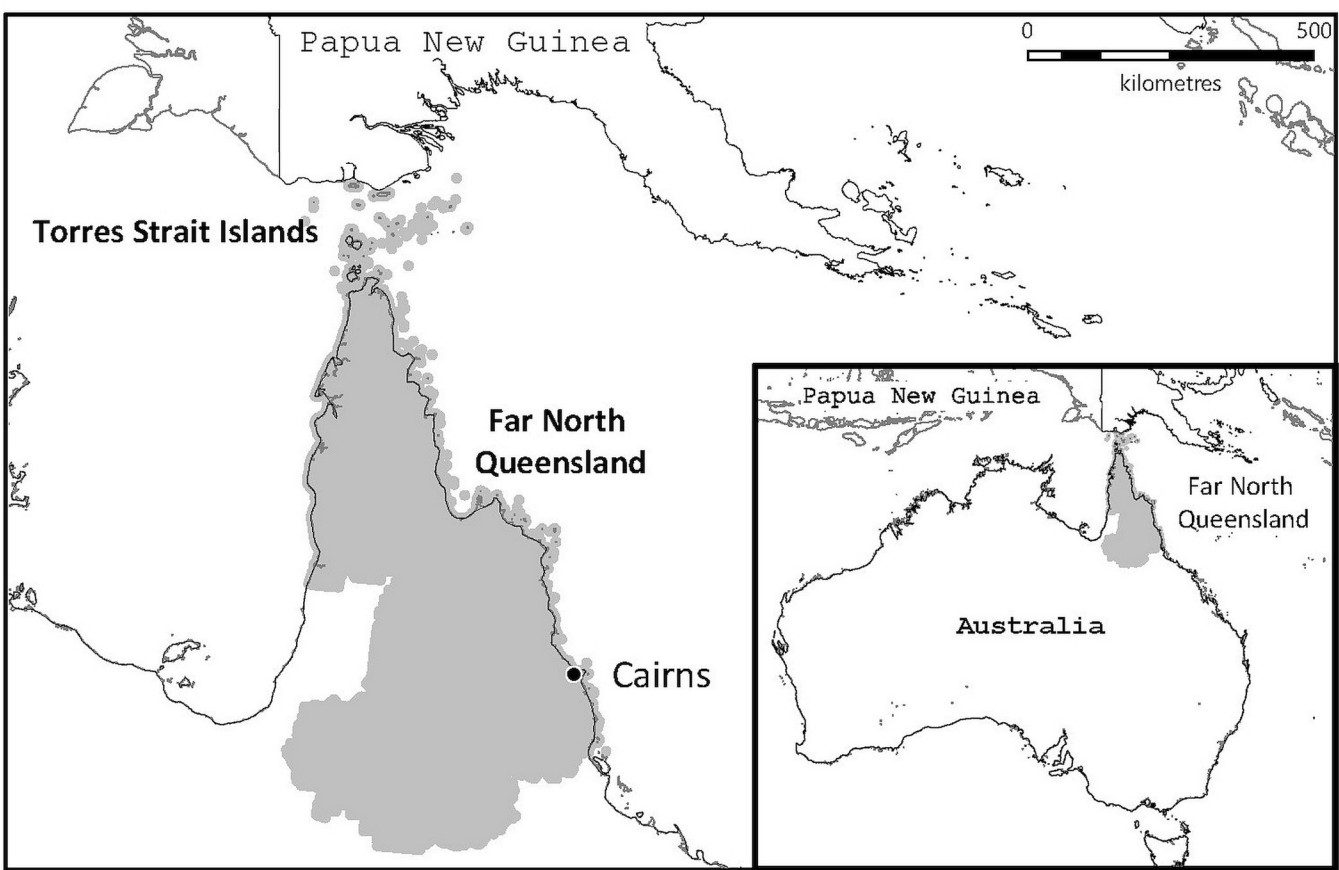

**Fig 1. The study region of Far North Queensland in tropical Australia.** The map was constructed using mapping software (MapInfo version 15.02, Connecticut, USA) using data provided by the State of Queensland (QSpatial). Queensland Place Names—State of Queensland (Department of Natural Resources, Mines and Energy) 2019, available under Creative Commons Attribution 4.0 International licence https://creativecommons.org/licenses/by/4.0/. 'Coastline and state border–Queensland—State of Queensland (Department of Natural Resources, Mines and Energy) 2019, available under Creative Commons Attribution 4.0 International licence https://creativecommons.org/licenses/by/4.0/.

This study was performed to gain a greater understanding of the seasonality of these locally important tropical infections and identify the drivers of any observed seasonality. It was hoped that this would provide insights that may enhance the clinical care of patients with undifferentiated illness in the region and inform public health strategies to mitigate the local impact of these life-threatening pathogens.

## Materials and methods

Cairns Hospital is the sole tertiary referral hospital and public microbiology service provider for FNQ. All laboratory confirmed cases of melioidosis, leptospirosis, rickettsial infection, cryptococcal infection, influenza virus infection and *S. aureus* bloodstream infection (SABSI) managed within the Cairns and Hinterland Health Service (CHHHS) between January 1998 and December 2019 were included. Melioidosis, leptospirosis, rickettsial diseases and cryptococcal infection are the most common tropical pathogens in the region, while influenza and SABSI were chosen as comparators; influenza is the archetypal seasonal disease, while seasonality has also been reported in *S. aureus* infection [8, 33]. The study period was chosen as it coincided with the introduction of a statewide electronic pathology system.

### Data collection

Participants were identified using AUSLAB, the electronic pathology and forensic laboratory system that serves the needs of the Queensland Health public health system. Melioidosis, leptospirosis and rickettsial cases were identified as described previously [27–30]. A single blood culture positive for *S. aureus* defined SABSI. Influenza cases were identified using the Tropical Public Health Unit database. This was restricted to data from 2001—with only minimal data between 2001–2008—as the database became established and molecular testing was introduced. Weather data were obtained from the Australian Bureau of Meteorology (BoM). This was recorded at Cairns Aero Station, located in Cairns City approximately 7km from Cairns Hospital. A cyclone was included if it passed within 200km of Cairns City. Mild, moderate, or severe flooding as defined by the BoM within the region's three major river systems–the Mulgrave-Russell basin, the Daintree basin and the Herbert-Johnstone basins–were identified from BoM records [34]. The wet season was defined as the months November-April (inclusive) with the dry season defined as May-October (inclusive). Census data from the Australian Bureau of Statistics were used for incidence calculations [16].

### Statistical analysis

Data were de-identified, entered into an electronic database (S1 Dataset, Microsoft Excel 2016, Microsoft, Redmond, WA, USA) and analysed using statistical software (Stata version 14.2, StataCorp LLC, College Station, TX, USA). The statistical significance of trends over time were assessed using an extension of the Wilcoxon rank-sum test [35]. Climatic factors were arranged into groups and their association with infection data was assessed using Poisson regression analysis. As many variables had a non-parametric distribution, groups were compared using the Kruskal-Wallis test. Correlation coefficients between continuous variables were determined using Spearman's rho. Multivariate analyses were not performed since all climate variables except for cloud cover were strongly correlated (S1 Table). P-values <0.05 were considered statistically significant.

### Ethical approval

The Far North Queensland Human Research Ethics Committee provided ethical approval for the study (HREC/2020/QCH/67479–1470) including access to previously approved databases (HREC/16/QCH/110–1083, HREC/15/QCH/46, HREC/16/QCH/37–1043 LR, HREC/17/QCH/66–1148 QA). As the data were retrospective and de-identified, the Committee waived the requirement for informed consent.

## Results

Individual climate factors remained stable during the study period; the small increase in mean temperature, mean cloud cover and mean dew point did not reach statistical significance (Table 1). Fourteen cyclones came within 200km of Cairns city and there were 55 local flooding events, but neither cyclones (p for trend = 0.81) nor flooding events (p for trend = 0.31) showed an increase in frequency during the study period. Between January 1998 and December 2019, the population of FNQ increased by 32% from 217,377 to 286,995 [16].

There were 297 cases of melioidosis during the study period: the annual incidence increased from 4.6/100,000 population in 1998 to 9.8/100,000 population in 2019 (p<0.001) (Fig 2). Infections were seasonal with 232 (78.1%) cases diagnosed during the wet season (p<0.001) (Fig 3). The number of monthly melioidosis cases was associated with mean rainfall, mean temperature, mean humidity, mean cloud cover, and mean dew point at time of diagnosis.

**Table 1. Changes in climatic factors, Cairns, Far North Queensland, 1998–2019.**

|  | 1998–2002 | 2003–2007 | 2008–2011 | 2012–2015 | 2016–2019 | p [a] |
|---|---|---|---|---|---|---|
| Mean (95% CI) monthly rainfall (mm) | 164 (103–224) | 160 (104–217) | 202 (129–275) | 146 (94–198) | 157 (93–220) | 0.61 |
| Mean (95% CI) temperature (˚Celsius) | 29.4 (28.9–29.9) | 29.2 (28.7–29.8) | 29.5 (28.9–30.0) | 29.5 (28.8–30.1) | 29.9 (29.2–30.5) | 0.32 |
| Mean (95% CI) cloud cover (oktas) | 4.6 (4.1–5.0) | 4.5 (4.0–4.9) | 4.7 (4.2–5.1) | 3.7 (3.2–4.1) | 5.2 (4.3–6.1) [b] | 0.13 |
| Mean (95% CI) 9 AM humidity (%) | 70.2 (68.0–72.4) | 70.3 (68.2–72.5) | 70.4 (68.1–72.6) | 66.2 (64.1–68.3) | 70.5 (67.9–73.0) | 0.18 |
| Mean (95% CI) 9AM dew point (˚Celsius) | 19.3 (18.4–20.3) | 19.7 (18.9–20.5) | 19.8 (18.9–20.7) | 18.9 (17.9–19.8) | 20.1 (19.1–21.0) | 0.93 |
| Number of severe weather events [c] | 20 | 18 | 9 | 16 | 8 | 0.77 |

CI, confidence interval.

[a] p for trend value calculated using annual data.

[b] Some cloud cover data were not available for the years 2017–2019.

[c] Cyclone within 200km of Cairns and/or flood within the Mulgrave-Russell, Daintree and or Herbert-Johnstone basins.

Case numbers were higher during months that contained a severe weather event (Tables 2 and 3).

There were 406 cases of leptospirosis during the study period, but no change in annual incidence (p = 0.37) (Fig 2). Infections were seasonal with 286 (70.4%) diagnosed in the wet season (p<0.001) (Fig 3). The number of monthly leptospirosis cases were associated with mean rainfall, mean temperature, mean humidity, mean cloud cover, and mean dew point at time of diagnosis. Cases were more common in months with a severe weather event (Tables 2 and 3).

There were 160 cases of rickettsial disease which included 114 (71%) cases of scrub typhus and 43 (27%) cases of QTT; three cases were unable to be speciated. The incidence of rickettsial infections increased during the study period (nadir of 0.4/100,000 population in 2006 a peak of 8.0/100,000 population in 2014 (p = 0.008) (Fig 2). There was no seasonal trend overall, but subgroup analyses showed QTT infections were more common in the drier autumn and winter (Fig 3, S1 Fig). The two infections had no association with individual climatic factors or severe weather events (Tables 2 and 3).

There were 49 cases of cryptococcosis, speciation was possible in 32/49 (65.3%); *C. neoformans* accounted for 17/32 (53.1%) and *C. gattii* 15/32 (46.9%). The mean (95% CI) incidence of all cryptococcal disease during the study period was 8.7/1,000,000 population and the mean (95% CI) incidence of culture-confirmed *C. gattii* was 2.8/1,000,000 (1.1–4.5) population (Fig 2). While cases occurred throughout the year, the majority of confirmed *C. gattii* (11/15, 73%) presented during the drier months of May-November with a spike in the month of July (Fig 3). There was no association between the incidence of cryptococcal infections and individual climatic factors or severe weather events (Tables 2 and 3).

There were 2174 cases of SABSI over the study period: 1692 (78%) were methicillin-sensitive *Staphylococcus aureus* (MSSA) and 482 (22%) were MRSA. The mean incidence of SABSI was 42.4/100,000 population (MSSA 32.3/100,000, MRSA 9.1/100,000) which remained unchanged during the study period (p = 0.87) (Fig 2). Infections were not seasonal and were seen throughout both the wet and dry seasons (Fig 3). There was no association between SABSI and severe weather events or individual climatic factors (Tables 2 and 3).

There were 11579 confirmed cases of influenza between 2001 and 2019, although this is certainly an underestimate as diagnostic strategies evolved during the study period. Case numbers fluctuated with large numbers seen during the 2009 H1N1 epidemic and throughout 2016 and 2018. Cases occurred predominantly in the winter-spring months but were seen throughout the year in most years (Fig 2). Overall, there were more cases in the cooler, dry season (p = 0.008) but there was an additional peak in the hot, wet months of Feb-March in some

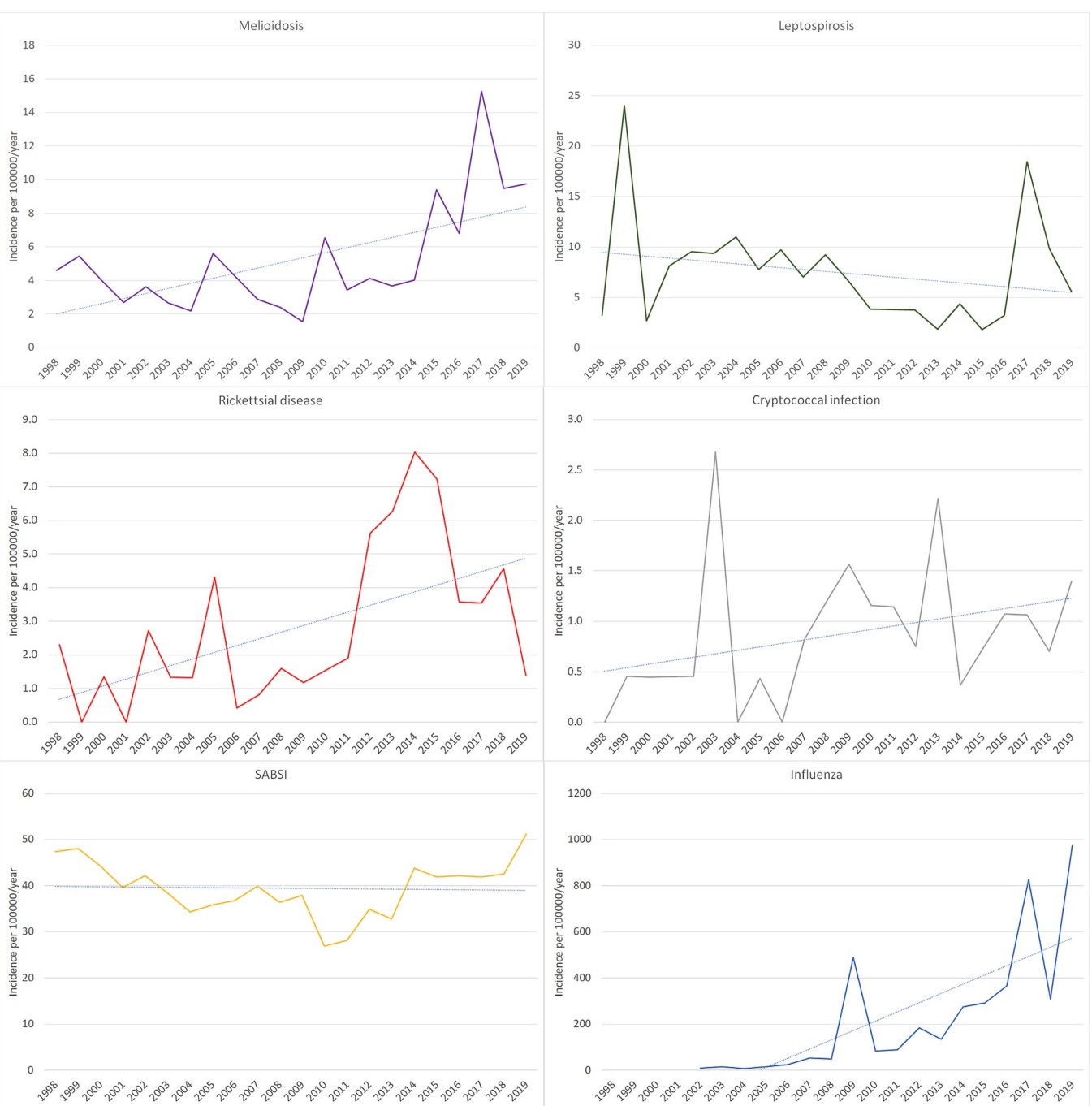

**Fig 2. Incidence of the infections in Cairns, Far North Queensland, 1998–2019.**

years (S2 Fig). All individual climate factors were associated with influenza cases but there was no association with severe weather events (Tables 2 and 3).

## Discussion

This study of seasonality of important infections in this region of tropical Australia provides an insight into the complex interplay between environmental conditions, urban expansion,

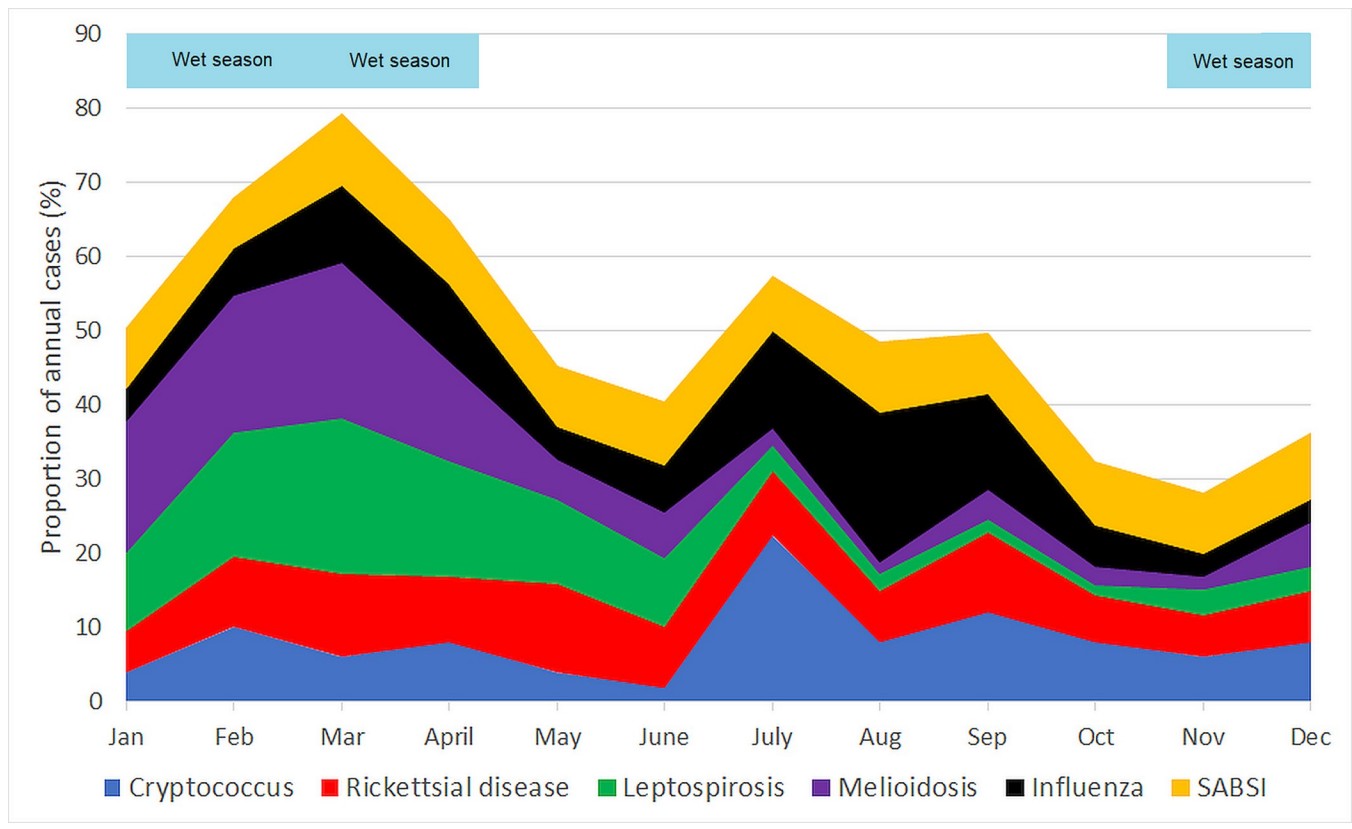

**Fig 3. Proportion of the annual number of cases seen in each month in Cairns, Far North Queensland, 1998–2019.** The November-April (inclusive) wet season is highlighted in pale blue at the top of the figure.

**Table 2. Association of individual climatic factors with the different infections in Far North Queensland, 1998–2019.**

| Infection | Rainfall [a] | | Temperature [b] | | Cloud cover [c] | | Humidity [d] | | Dew point [e] | |
|---|---|---|---|---|---|---|---|---|---|---|
| | IRR | p | IRR | p | IRR | p | IRR | p | IRR | p |
| Melioidosis | 1.21 (1.17–1.25) | <0.001 | 1.27 (1.19–1.35) | <0.001 | 1.24 (1.14–1.35) | <0.001 | 1.33 (1.24–1.43) | <0.001 | 2.00 (1.77–2.27) | <0.001 |
| Leptospirosis | 1.14 (1.11–1.18) | <0.001 | 1.10 (1.05–1.16) | <0.001 | 1.13 (1.06–1.21) | <0.001 | 1.50 (1.41–1.60) | <0.001 | 1.89 (1.71–2.10) | <0.001 |
| Rickettsial disease | 1.01 (0.95–1.06) | 0.85 | 0.99 (0.91–1.06) | 0.70 | 0.83 (0.74–0.93) | 0.001 | 0.97 (0.88–1.07) | 0.54 | 0.99 (0.86–1.15) | 0.99 |
| Cryptococcosis | 0.95 (0.86–1.06) | 0.38 | 0.92 (0.80–1.05) | 0.21 | 0.99 (0.82–1.22) | 0.98 | 0.85 (0.71–1.01) | 0.06 | 0.85 (0.67–1.09) | 0.21 |
| SABSI | 0.99 (0.98–1.01) | 0.22 | 1.01 (0.99–1.03) | 0.16 | 1.01 (0.98–1.04) | 0.59 | 0.99 (0.97–1.02) | 0.77 | 1.00 (0.97–1.04) | 0.94 |
| Influenza | 0.94 (0.93–0.94) | <0.001 | 0.90 (0.89–0.91) | <0.001 | 0.99 (0.97–0.99) | 0.046 | 1.08 (1.07–1.10) | <0.001 | 0.84 (0.83–0.85) | <0.001 |

Poisson regression presented

IRR: incidence rate ratio

SABSI: *Staphyloccocus aureus* bloodstream infection

[a] 9 groups based on total monthly rainfall: 0-49ml, 50-199ml, 100-149ml, 150-199ml, 200-249ml, 250-299ml, 300-349ml, 350-399ml, ≥400ml

[b] 10 groups based on mean temperature: <25˚C, 25.0–25.9˚C, 26.0–26.9˚C, 27.0–27.9˚C, 28.0–28.9˚C, 29.0–29.9˚C, 30.0–30.9˚C, 31.0–31.9˚C, 32.0–32.9˚C, ≥33.0˚C

[c] 8 groups based on mean okta

[d] 8 groups based on mean humidity at 0900: 50.0–54.9%, 55.0–59.9%, 60.0–64.9%, 65.0–69.9%, 70.0–74.9%, 75.0–79.9%, 80.0–84.9%, 85.0–90.0%

[e] 6 groups based on dewpoint at 0900: 9.0–11.9˚C, 12.0–14.9˚C, 15.0–17.9˚C, 18.0–20.9˚C, 21.0–23.9˚C, ≥24.0˚C

**Table 3. Infection numbers associated with severe weather events, in Far North Queensland, 1998–2019.**

| | Cases in months with a cyclone or flood [a] n = 39 | Cases in months with neither cyclone nor flood [a] n = 225 | p |
|---|---|---|---|
| Melioidosis | 2 (0–3) | 0 (0–1) | p<0.001 |
| Leptospirosis | 2 (0–4) | 1 (0–2) | 0.001 |
| Rickettsial disease | 0 (0–1) | 0 (0–1) | 0.68 |
| Cryptococcosis | 0 (0–0) | 0 (0–0) | 0.87 |
| SABSI | 10 (6–11) | 9 (6–11) | 0.98 |
| Influenza | 7 (0–50) | 17 (2–61) | 0.23 |

All numbers presented are medians (interquartile range)

[a] Cyclone within 200km of Cairns and/or flood within the Mulgrave-Russell, Daintree and or Herbert-Johnstone basins.

SABSI: *Staphyloccocus aureus* bloodstream infection

host behaviour, and the characteristics of individual pathogens. The climate was stable over the course of the 21-year study period, but the incidence of melioidosis and rickettsial diseases increased significantly, reinforcing the fact that factors other than the prevailing climate frequently have a greater impact on the burden of these life-threatening infections. Each of the infections studied in this report have been previously linked to individual climatic factors [6–11, 36]. However, the data in this study—collected over a 21-year period—highlight that weather alone cannot explain either observed seasonality or infection trends more generally. Instead, other factors including host behaviour, host immune function, urban planning and societal inequality also need to be considered [1].

Melioidosis is inherently seasonal, seen predominantly during the wet season, and this study's findings reinforce this [37]. The association with dew point, cloud cover and rainfall reflects the optimal conditions for the growth and survival of the causative organism, *Burkholderia pseudomallei* [38]. Rainfall also disrupts topsoil, increasing the potential for the pathogen to interact with the host. Although more cases were seen in months where there was a cyclone or flood than in those without, these severe weather events were infrequent and their independent contribution to disease incidence is confounded by the fact that 95% of these events occurred during the wet season, when their occurrence is already greater [39]. However, the rise in the incidence of melioidosis seen in this study cannot be explained by weather patterns —which were relatively stable over the course of the study—and there is no evidence of increased case finding in the region [40]. Instead, urban expansion resulting in major infrastructure development with associated disruption of topsoil is more likely to explain incidence doubling over the study period [40, 41]. The Cairns city region experienced a greater than 10-fold increase in the incidence of melioidosis during the study period and this has been hypothesised to be linked to the construction of a large motorway to the south of the city [40].

The FNQ population also has high rates of co-morbidities that increase the risk of melioidosis—particularly diabetes mellitus and hazardous alcohol use–especially among disadvantaged, predominantly Indigenous, populations living in remote communities [24]. The strong association between socioeconomic disadvantage and the incidence—and even outcome—of melioidosis emphasises that although the disease is seasonal, it is host—and societal—factors that play a far greater role in the burden of this opportunistic pathogen [24, 42]. However, addressing these factors is not easy: public health strategies that focus on human behaviour and that aim to prevent exposure to *B. pseudomallei* are challenging to implement [43–45]. Meanwhile, chemoprophylaxis would not be cost-effective locally and is associated with significant side effects [46].

Leptospirosis is also a wet season disease that has been linked with rainfall and flooding events; an observation which was replicated in this study [47]. The association between cases

and severe weather event is again confounded by the fact that these events usually occur in the wet season when case numbers are greatest. Leptospires survive for long periods of time in freshwater and survival may be further prolonged by warmer temperatures [48]. The association with dew point and cloud cover may also support the hypothesis that soil acts as a reservoir of leptospires, although confirmation of this hypothesis would require a more detailed environmental study [49].

The incidence of leptospirosis was stable during the study period despite regular public awareness campaigns that aimed to prevent infection [50]. As approximately 85% of leptospirosis cases occurred in patients in whom occupational exposure, particularly banana farming, may have been a factor, this may reflect difficulty in targeting the seasonal workers that comprise a significant proportion of this workforce [27, 51]. It might be hypothesised that chemoprophylaxis might have utility in these populations, however, given the potential side effects associated with doxycycline—particularly photosensitivity in a tropical environment—and the very low local case-fatality rates—chemoprophylaxis is unlikely to be accepted by local workers [27, 52, 53].

The tenfold increase in rickettsial infections despite a stable climate suggests a role for other explanations for the rising disease incidence. Increased case finding is unlikely to account for this; the same molecular diagnostics have been used since 2009 with the largest increase in cases occurring from 2012 onwards. Population growth leading to expansion of the urban-rural fringe is likely to have increased interactions with tick and mite habitats. As the climate changes, it is possible that with the predicted increase in temperatures, the range and distribution of vertebrate hosts carrying infected mites and ticks may change, which may have an additional impact on disease incidence [54].

The lack of seasonality of scrub typhus in our study is similar to findings in tropical Vietnam and Thailand that contrasts with the summertime peaks in northern Asia [10]. This is likely to reflect year-round mite activity in tropical regions. Rainfall was linked to infection in a previous small study in the Torres Strait Islands, a region in the north of FNQ, but this finding was not replicated in our larger cohort [55]. Although QTT infections occurred year-round, there was a trend to predominance in the dry season as has been previously documented in subtropical Australia [56]. This likely reflects increased tick density and activity during the tropical winter months, but also, probably more importantly, the increased popularity of outdoor activities such as camping and hiking at this time, increasing the risk of tick exposure [57].

The rate of cryptococcal disease is higher than in other areas in Australia (mean incidence: 2.62 per million compared to a national figure of 0.6 per million) [30, 58]. This is likely to be driven by the higher proportion of *C. gattii* infections among local patients with cryptococcal infection. Almost 90% of these local *C. gattii* infections occurred in residents living in remote, rural locations where *Eucalyptus camaldulensis* and *E. tereticornis* trees—an environmental niche for *C. gattii* in Australia—are abundant [30, 59, 60]. The flowering of *E. camaldulensis* has been linked to cryptococcal infection, possibly promoting airborne dispersal of infectious propagules which might then be inhaled [61]. Most cases of confirmed *C. gattii* in this cohort presented during winter-spring (June-November) with a small spike in July. *E. camaldulensis* and *E. tereticornis* exhibit seasonal flowering predominantly in FNQ in June-August, at a time when the region has less rainfall, cooler temperatures and increased mean wind speeds [62–64]. However, the incubation period of *C. gattii* is uncertain and inoculation events are difficult to define, so an association between Eucalypt flowering and *C. gattii* infection remains only a hypothesis at this stage [60].

The cohort had a very high incidence of SABSI—nearly 20% greater than the previously estimated nationwide Australian annual incidence of 35/100,000 population—but it remained

stable over the course of the study [65]. Although the proportion of MRSA among local *S. aureus* isolates is one of the highest in Australia, and rising, it was notable that the proportion of SABSI due to MRSA did not increase [31]. The lack of SABSI seasonality or any association with individual climate factors in this cohort may suggest that climate factors have a greater impact on skin and soft tissue infection than on invasive infections like SABSI [8]. It also likely reflects the perennial role of other influences, particularly socioeconomic factors. Indigenous Australians suffer disproportionately from *S. aureus* infection in the region, and this is linked to the ongoing, substantial socioeconomic disadvantage experienced in many members of these communities [31, 66].

Diagnostic methods for influenza evolved significantly during the study period precluding analysis of long-term trends in disease incidence. However, it was still possible to examine seasonal variation and the inter-seasonality seen in this analysis is consistent with other studies [32, 67]. The second peak in February-March during most years between 2014 and 2019 has been noted in other tropical and subtropical regions in Australia (S2 Fig) [68]. This may reflect higher humidity and rainfall during these months promoting virus survival or the persistence of specific strains of influenza [69]. Year-round tourism with importation of northern hemisphere virus during the Australian summer may also contribute [70]. These findings underscore the need for year-round influenza testing and surveillance. It may also influence vaccination strategies, with biannual vaccination a potential option in vulnerable populations [71].

For clinicians working in the region, the study highlights that melioidosis and leptospirosis are strongly seasonal, with both diagnoses requiring consideration in the work-up of patients with an acute, undifferentiated presentation in the wet season. Early recognition of these pathogens is crucial because both can be rapidly fatal in the absence of prompt antimicrobial therapy and appropriate supportive care [72]. However, in contrast, the lack of clear seasonality in the presentation of scrub typhus, cryptococcosis and SABSI in the region means that clinicians need to consider the possibility of these infections year-round. This is important, because scrub typhus and cryptococcus require specific diagnostic testing and therapies which are not necessarily included in typical empirical regimens; again, both conditions can cause life-threatening disease, even in well-resourced settings [30, 73].

Our study has several important limitations. The retrospective and observational nature means that the data are incomplete and, while associations can be identified, causality cannot be established. Rainfall, temperature, humidity, and dewpoint data were, as would be expected, highly correlated, precluding reliable multivariate analysis of these covariates. Accordingly, our analyses cannot identify which of these climactic variables is most causally related to the incidence of the examined infections. In addition, BoM cloud cover data between 2017–19 were incomplete, however, this is not thought to have impacted the findings substantially. The incidence of the infections will be underestimated as only cases hospitalised in the public health system were included; private laboratory and health system data were not available. However, this would not be expected to impact on the seasonality identified and the observed trends in incidence. Expansion of the local infectious diseases service, improvements in diagnostic testing and increased awareness may have increased case finding throughout the period. Overall, blood culture testing per head of population did increase during the study period (p for trend p<0.001) which may have contributed to increased SABSI and melioidosis case numbers however there has been no change in disease severity of melioid to suggest ascertainment bias [40]. Molecular diagnostic testing for rickettsial disease and leptospirosis evolved during the study period, which have increased the number of confirmed diagnoses, however the vast majority of rickettsial infections were diagnosed serologically so this is thought to have only a minor influence [29]. Scrub typhus and QTT are presented together despite being distinct

infections—and some may take issue with this approach—however, the similarities in the diseases, we felt, justified a pragmatic approach. SABSI were not divided into hospital or community-acquired infections however previous studies have demonstrated that the majority are community-acquired [74]. Limited influenza data precluded analysis of longitudinal trends although our study clearly highlights the need for perennial monitoring for influenza. Finally, caution should be taken in extrapolating seasonal trends found in this study to other tropical regions. The multicultural population of FNQ lives in a unique and remote part of Australia that contains rainforest, marine environments, lush arable land, and dry savannah. This study has highlighted the primacy of human and environmental factors in the incidence of several of the examined infections; it therefore emphasises the need for local population and environmental data in the evaluation of seasonality of these infections in other regions.

## Conclusions

The FNQ region is vulnerable to anthropogenic climate change but while the burden of several of the infections rose during the study period, the climate did not alter significantly. Instead, urban expansion, socioeconomic inequality, and the way that humans interact with their environment appeared to have a far greater impact on the incidence of these diseases during the study period. While seasonal variation in the presentation of some pathogens is noted, and climate change has the potential to influence this still further, it remains essential to also consider the broader anthropological and environmental drivers of infection if the burden of these pathogens is to be mitigated.

## Supporting information

**S1 Fig. Monthly cases of rickettsial infections by species, Cairns, Far North Queensland, 1998–2019.**
(TIF)

**S2 Fig. Influenza cases by month, Cairns, Far North Queensland, 2009–2019.** Only data after 2008 are presented as data collection prior to 2009 were incomplete.
(TIF)

**S1 Table. Correlation between individual climatic variables, Cairns, Far North Queensland, 2001–2019.**
(DOCX)

**S1 Dataset.**
(XLSX)

## Acknowledgments

The authors would like to thank Sally Rubenach, epidemiologist and Manager of Health Surveillance at the Tropical Public Health Services Cairns, for assistance with influenza data. They would like to thank Peter Horne for his assistance with the preparation of Fig 1. They would also like to thank Matthew Barrett at Australian Tropical Herbarium, James Cook University for assistance with accessing the Atlas of Living Australia.

## Author Contributions

**Conceptualization:** Lee J. Fairhead, Simon Smith, Josh Hanson.

**Data curation:** Lee J. Fairhead, Simon Smith, Beatrice Z. Sim, Alexandra G. A. Stewart, James D. Stewart.

**Formal analysis:** Matthew Law, Josh Hanson.

**Investigation:** Lee J. Fairhead, Simon Smith, Beatrice Z. Sim, Alexandra G. A. Stewart, James D. Stewart, Matthew Law, Josh Hanson.

**Methodology:** Lee J. Fairhead, Simon Smith, Josh Hanson.

**Supervision:** Simon Smith, Enzo Binotto, Matthew Law, Josh Hanson.

**Validation:** Josh Hanson.

**Visualization:** Simon Smith, Josh Hanson.

**Writing – original draft:** Lee J. Fairhead.

**Writing – review & editing:** Lee J. Fairhead, Simon Smith, Beatrice Z. Sim, Alexandra G. A. Stewart, James D. Stewart, Enzo Binotto, Matthew Law, Josh Hanson.

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
