## [Decision Letter · Decision Letter 0]

25 Mar 2022

PGPH-D-22-00139

The seasonality of infections in tropical Far North Queensland, Australia: a 21-year retrospective evaluation of the seasonal patterns of six endemic pathogens

Dear Dr. Hanson,

Thank you for submitting your manuscript to PLOS Global Public Health. After careful consideration, we feel that it has merit but does not fully meet PLOS Global Public Health’s publication criteria as it currently stands. Therefore, we invite you to submit a revised version of the manuscript that addresses the points raised during the review process.

We look forward to receiving your revised manuscript.

Kind regards,

Karen D. Cowgill, PhD, MSc

Academic Editor

Journal Requirements:

1. Please update your Competing Interests statement. If you have no competing interests to declare, please state: “The authors have declared that no competing interests exist.”

Additional Editor Comments (if provided):

Reviewers' comments:

Reviewer's Responses to Questions

**Comments to the Author**

1. Does this manuscript meet PLOS Global Public Health’s publication criteria? Is the manuscript technically sound, and do the data support the conclusions? The manuscript must describe methodologically and ethically rigorous research with conclusions that are appropriately drawn based on the data presented.

Reviewer #1: Yes

Reviewer #2: Yes

2. Has the statistical analysis been performed appropriately and rigorously?

Reviewer #1: Yes

Reviewer #2: I don't know

3. Have the authors made all data underlying the findings in their manuscript fully available (please refer to the Data Availability Statement at the start of the manuscript PDF file)?

Reviewer #1: Yes

Reviewer #2: Yes

4. Is the manuscript presented in an intelligible fashion and written in standard English?

Reviewer #1: Yes

Reviewer #2: Yes

5. Review Comments to the Author

Reviewer #1: This study examined seasonality with data accumulated over a long period of 21 years. Although it shows very interesting results, the logical development to draw conclusions is insufficient. It is recommended to supplement this point and reflect it in the mabuscript. On the other hand, the manuscript is very well written in an easy to read English style.

Lines 37-39 (Abstract) : Seasonality is most affected by temperature and humidity. And the most important factors that determine climate are temperature and humidity. Therefore, it is necessary to define specifically what environmental, human, and climatic factors mean.

(Introduction) The meaning of seasonality, the characteristics of the study region, the characteristics of the diseases to be studied, and why this diseases were chosen should be logically developed.

Lines 99-102 (Introduction) This part should be mentioned in the discussion or conclusion.

Lines 108-109 (Materials and methods) Why this disease was selected to be studied, should be dealt with in the introduction section.

Lines 110-111, & 113 (Materials and methods) A detailed description of this database 'Queensland electronic pathology database' should be added.

Lines 129-138 (Statistical Analysis) As a whole, non-parametric analysis methods were applied. Why?

Lines 135-136 (Statistical Analysis) Spearman's rho is a calculation method for estimating the correlation coefficient. What do you mean by used to compare continuous variables?

Lines 151-152 (Results) " but neither ~ respectively) Please state in a readable way that it is not statistically significant.

Lines 153, 215, 221 Please be consistent with putting a thousandth comma in a number.

Lines 162-163 (Fig 2) There should be an indication for the wet season in Fig 2.

Page 9, Table 2 It seems that the 95% confidence interval in parentheses is, but it is more readable to express it by changing the line rather than spreading it over two lines.

Pages 229-265 (Discussion) It is not necessary to list by pathogen in the discussion. The text should be written more condense with the main points.

Reviewer #2: "There were 406 cases of leptospirosis during the study period, but no change in annual 194 incidence (p=0.37) (Fig 1). Infections were seasonal with 286 (70.4%) diagnosed in the wet

195 season (p=0.0001) (Fig 2) "

- It is better to report P values smaller than 0.001 as P<0.001

6. PLOS authors have the option to publish the peer review history of their article (what does this mean?). If published, this will include your full peer review and any attached files.

**Do you want your identity to be public for this peer review?** For information about this choice, including consent withdrawal, please see our Privacy Policy.

Reviewer #1: No

Reviewer #2: No

---

## [Decision Letter · Decision Letter 1]

1 May 2022

The seasonality of infections in tropical Far North Queensland, Australia: a 21-year retrospective evaluation of the seasonal patterns of six endemic pathogens

PGPH-D-22-00139R1

Dear Dr Hanson,

We are pleased to inform you that your manuscript 'The seasonality of infections in tropical Far North Queensland, Australia: a 21-year retrospective evaluation of the seasonal patterns of six endemic pathogens' has been provisionally accepted for publication in PLOS Global Public Health.

Before your manuscript can be formally accepted you will need to complete some formatting changes, which you will receive in a follow up email. A member of our team will be in touch with a set of requests. You may also wish to take into account the feedback from Reviewer #1 (see below).

Best regards,

Karen D. Cowgill, PhD, MSc

Academic Editor

Reviewer Comments (if any, and for reference):

Reviewer's Responses to Questions

**Comments to the Author**

1. If the authors have adequately addressed your comments raised in a previous round of review and you feel that this manuscript is now acceptable for publication, you may indicate that here to bypass the “Comments to the Author” section, enter your conflict of interest statement in the “Confidential to Editor” section, and submit your "Accept" recommendation.

Reviewer #1: All comments have been addressed

Reviewer #2: All comments have been addressed

2. Does this manuscript meet PLOS Global Public Health’s publication criteria? Is the manuscript technically sound, and do the data support the conclusions? The manuscript must describe methodologically and ethically rigorous research with conclusions that are appropriately drawn based on the data presented.

Reviewer #1: Yes

Reviewer #2: Yes

3. Has the statistical analysis been performed appropriately and rigorously?

Reviewer #1: Yes

Reviewer #2: I don't know

4. Have the authors made all data underlying the findings in their manuscript fully available (please refer to the Data Availability Statement at the start of the manuscript PDF file)?

Reviewer #1: Yes

Reviewer #2: Yes

5. Is the manuscript presented in an intelligible fashion and written in standard English?

Reviewer #1: Yes

Reviewer #2: Yes

6. Review Comments to the Author

Reviewer #1: I still think the Discussion section is too long and not condense. Why don't the authors explain the key points as simply and clearly as you answered at the "Response to reviewers"?

Reviewer #2: no further comments

7. PLOS authors have the option to publish the peer review history of their article (what does this mean?). If published, this will include your full peer review and any attached files.

**Do you want your identity to be public for this peer review?** For information about this choice, including consent withdrawal, please see our Privacy Policy.

Reviewer #1: No

Reviewer #2: No
